# Habitual coffee consumption poorly correlates with sleep quality and daytime sleepiness: A cross-sectional study

Simon Söderholm[1,2]*, Martin Ulander[3,4], Vanessa William Toma[3], Sara Kaufmann[5], Xiangyu Qiao[3], Daniel Berglind[6,7,8], Susanna Calling[9,10], Bledar Daka[11], Ludger Grote[12,13], Mats Martinell[14], Frida Bergman[15], Pontus Henriksson[16], Carl-Johan Östgren[5,17], Wen Zhong[3,18], Claudio Cantù[1,2]*, Fredrik Iredahl[1,5]*

1 Wallenberg Centre for Molecular Medicine, Linköping University, Linköping, Sweden, 2 Department of Biomedical and Clinical Sciences, Division of Molecular Medicine and Virology, Faculty of Medicine and Health Sciences, Linköping University, Linköping, Sweden, 3 Department of Biomedical and Clinical Sciences, Division of Cell and Neurobiology, Faculty of Medicine and Health Sciences, Linköping University, Linköping, Sweden, 4 Department of Clinical Neurophysiology, Linköping University Hospital, Linköping, Sweden, 5 Primary Health Care Center, Department of Health, Medicine and Caring Sciences, Faculty of Medicine and Health Sciences, Linköping University, Linköping, Sweden, 6 Department of Biomedicine, Aarhus University, Aarhus, Denmark, 7 Centre for Epidemiology and Community Medicine, Region Stockholm, Stockholm, Sweden, 8 Center for Wellbeing, Welfare and Happiness, Stockholm School of Economics, Stockholm, Sweden, 9 Center for Primary Health Care Research, Department of Clinical Sciences, Lund University, Malmö, Sweden, 10 Primary care Skåne, Region Skåne, Kristianstad, Sweden, 11 Family medicine, School of Public Health and Community Medicine, Institute of Medicine, Sahlgrenska Academy, University of Gothenburg, Gothenburg, Sweden, 12 Center for Sleep and Vigilance Disorders, Institute of Medicine, Sahlgrenska Academy, University of Gothenburg, Gothenburg, Sweden, 13 Center for Sleep Medicine, Department of Respiratory Medicine, Sahlgrenska University Hospital, Gothenburg, Sweden, 14 Department of Public Health and Caring Sciences, Uppsala University, Uppsala, Sweden, 15 Department of Public Health and Clinical Medicine, Umeå University, Umeå, Sweden, 16 Department of Health, Medicine and Caring Sciences, Linköping University, Linköping, Sweden, 17 Centre of Medical Image Science and Visualization, Linköping University, Linköping, Sweden, 18 Department of Neuroscience, Karolinska Institute, Stockholm, Sweden

* Fredrik.iredahl@liu.se (FI); Claudio.cantu@liu.se (CC); simon.soderholm@liu.se (SS)

## Abstract

Coffee is the most common drink in the world, second only to water. This makes caffeine, the ingredient of coffee known for its wakefulness-promoting effects, one of the most used psychoactive substances. The psychoactive property of caffeine is well-characterized, and entails its interaction with the adenosine receptors, involved in sleep regulation. While studies have shown a deleterious immediate effect of caffeine on sleep, less is known about the effects of chronic caffeine exposure. In the present cross-sectional study, we investigated this relationship across a large cohort of 30,154 individuals participating in the Swedish Cardiopulmonary Bioimage Study (SCAPIS), which allowed us to compare habitual coffee intake with sleep habits, subjective estimate of daytime sleepiness, and underlying genetic variants. According to our analyses, different degrees of coffee consumption, confirmed by statistical association with previously reported genetic variants, showed very low association with estimated patterns of sleep habits or perceived daytime sleepiness. These results

**Data availability statement:** For access to pseudonymized SCAPIS phenotype data and genotype data, an ethical approval from the Swedish Ethical Review Board (Etikprövningsmyndigheten) is required, as well as an approval from the SCAPIS Data access board (https://www.scapis.org/data-access). SCAPIS data is available to be applied for by researchers based in Sweden or international researchers in collaboration with a researcher based in Sweden. Furthermore, regarding international data sharing, SCAPIS data is subject to confidentiality under Chapter 24, Section 8 of the Swedish Public Access to Information and Secrecy Act (OSL), and Section 7 of the Public Access to Information and Secrecy Ordinance. The analysis code used in this study is available at GitHub (https://github.com/SimonSoderholm/CoffeeSleep-Analysis) and archived on Zenodo (DOI: 10.5281/zenodo.18400906).

**Funding:** C.C is a fellow and F.I is an associated clinical fellow of Wallenberg Center for Molecular Medicine (WCMM) and receives financial support from the Knut and Alice Wallenberg Foundation. C.C is also supported by funding from the Swedish Research Council, Vetenskapsrådet (2021-03075 and 2023-01898) and Cancerfonden (CAN 2018/542 and 21 1572 Pj). This work was also financed by Dr P Håkanssons Foundation, and the SciLifeLab & Wallenberg Data Driven Life Science Program (KAW 2020.0239).

**Competing interests:** The authors have declared that no competing interests exist.

indicate that coffee may be less impactful on sleep habits than previously thought, or that other mechanisms, such as the adaptive capabilities of the adenosine system in adult coffee users, may dampen its psychoactive potency.

## Introduction

Throughout history, people of all cultures have consumed foods with psychoactive properties. Among these, coffee is currently the most common beverage in the world, second only to water [1].

Given the wakefulness-promoting effects of coffee, the association between its consumption and sleep patterns is widely studied [2]. A molecular mechanism for this effect has been found: caffeine, the principal psychoactive substance in coffee, is a central nervous system (CNS) stimulant which, together with theobromine and theophylline, belongs to a class of substances known as methylxanthines. Methylxanthines act as antagonists of the adenosine receptor [3]. Adenosine, together with its receptors ($A_1$, $A_{2A}$, $A_{2B}$ and $A_3$), are known to be involved in sleep homeostasis [4]. Caffeine has been proved to promote wakefulness primarily by blocking $A_{2A}$ [5].

In addition, genome-wide association studies (GWAS) have identified genetic predispositions that contribute to how individuals may be affected by caffeine. The most significantly associated variants are found to affect the Aryl-Hydrocarbon Receptor (*AHR*) and the Cytochrome P450 Family 1 Subfamily A Member 1 and 2 (*CYP1A1* and *CYP1A2*) genetic loci [6–8]. Consistently, the P450 system has a key role in the metabolism of coffee, with CYP1A2 constituting the primary metabolizer of caffeine [9–11]. AHR is involved in the positive transcriptional regulation of *CYP1A1* and *CYP1A2* [12,13]. [14]

While studies have shown that caffeine intake can impair subsequent sleep [14], less is known about the long-term effect of regular caffeine consumption.

Sweden is, together with its Scandinavian neighboring countries, one of the highest per capita coffee consumers [15]. This offers ideal large cohorts for the study of existing associations between coffee consumption and genetics or other lifestyle parameters. In the present cross-sectional study, we take advantage of the data deriving from a large cohort comprising 30,154 individuals aged between 50–64 participating in the Swedish Cardiopulmonary Bioimage Study (SCAPIS) [16] to investigate the association between habitual coffee consumption and sleep habits and daytime sleepiness.

## Results

### Coffee consumption in Swedish middle-aged individuals

A total of 25,381 participants were considered (Fig 1A, Table 1), of which 12,990 females (51%) and 12,391 males (49%). Data missingness was assessed with Little's MCAR test, which indicated that data were missing not completely at random (p < 0.001). However, as the proportion of missing data values only constitute about 3.4% across the entire dataset, and additional analyses including missingness

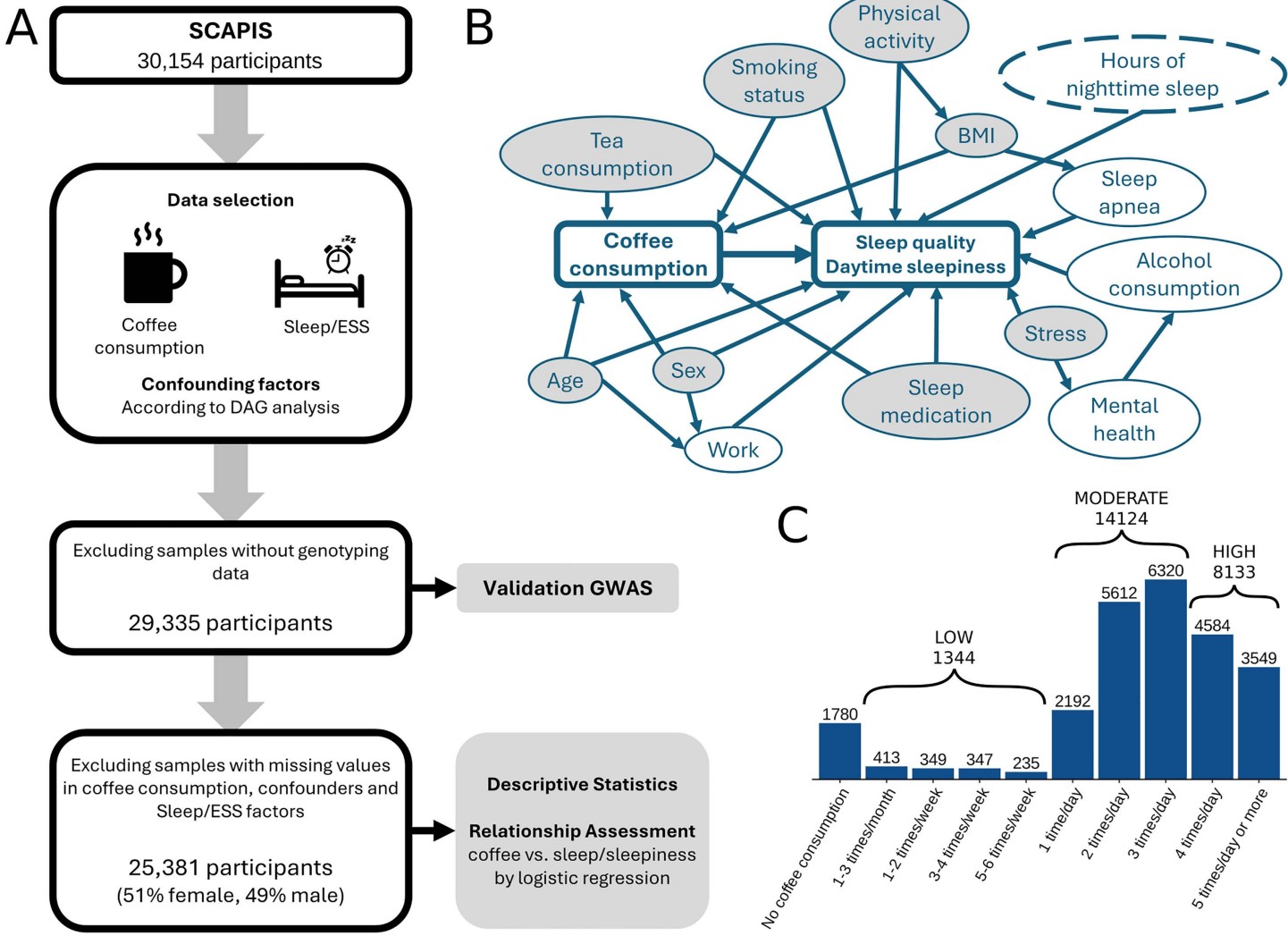

**Fig 1. Study overview. (A)** Schematic overview of the study, including data selection/filtration and main analysis steps. **(B)** Directed Acyclic Graph (DAG) that displays assumptions about the relationship between coffee consumption, sleep quality/ESS, and a selection of other variables. Potential confounders (i.e., factors that may influence both coffee consumption and sleep quality and therefore could bias the analysis) are indicated by a node with gray background color. **(C)** Overview of coffee consumption in the studied population. Bar plot showing the number of individuals (y-axis) for each level of coffee consumption (x-axis).

patterns and overimputation tests showed limited benefit of imputation (S1 Fig), we excluded participants with any missing data. The reported coffee consumption across the studied population has a distinct distribution (Fig 1C), where a majority of individuals have reported to drink coffee 1 or more times on a daily basis (n = 22,257, 87.7%), while fewer individuals have reported to only sporadically drink coffee monthly or weekly (n = 1344, 5.3%) or not drinking coffee at all (n = 1780, 7%).

From our directed acyclic graph (DAG) analysis (Fig 1B) we identified age, sex, body mass index (BMI), tea consumption, stress, physical activity, smoking status, and sleep medication use as relevant confounding factors. For daytime sleepiness, hours of nighttime sleep were defined as an additional confounding factor. Additional details on all variables used in the study, including coffee consumption, sleep/sleepiness, and confounding factors, are presented in S1 Table. Significant statistical association were found between each of the confounding factors and coffee consumption

**Table 1. Patterns of habitual coffee consumption by baseline characteristics.**

| Characteristic | Overall N = 25,381[1] | Coffee consumption | | | | p-value[2] |
| | | NONE N = 1,780[1] | LOW N = 1,344[1] | MODERATE N = 14,124[1] | HIGH N = 8,133[1] | |
|---|---|---|---|---|---|---|
| **Age** | 57.5 (4.3) | 56.6 (4.2) | 57.5 (4.3) | 57.8 (4.4) | 57.1 (4.3) | <0.001 |
| **Sex** | | | | | | <0.001 |
| Female | 12,990 (51%) | 1,093 (61%) | 633 (47%) | 8,129 (58%) | 3,135 (39%) | |
| Male | 12,391 (49%) | 687 (39%) | 711 (53%) | 5,995 (42%) | 4,998 (61%) | |
| **BMI** | 26.9 (4.4) | 26.9 (4.8) | 27.5 (4.9) | 26.7 (4.4) | 27.1 (4.1) | <0.001 |
| **Smoking Status** | | | | | | <0.001 |
| Never | 13,072 (52%) | 1,205 (68%) | 826 (61%) | 7,335 (52%) | 3,706 (46%) | |
| Ex-smoker | 9,315 (37%) | 452 (25%) | 410 (31%) | 5,369 (38%) | 3,084 (38%) | |
| Current | 2,994 (12%) | 123 (6.9%) | 108 (8.0%) | 1,420 (10%) | 1,343 (17%) | |
| **Tea consumption** | | | | | | <0.001 |
| NONE | 6,139 (24%) | 338 (19%) | 123 (9.2%) | 3,162 (22%) | 2,516 (31%) | |
| LOW | 10,397 (41%) | 303 (17%) | 392 (29%) | 5,783 (41%) | 3,919 (48%) | |
| MODERATE | 8,005 (32%) | 781 (44%) | 635 (47%) | 4,985 (35%) | 1,604 (20%) | |
| HIGH | 840 (3.3%) | 358 (20%) | 194 (14%) | 194 (1.4%) | 94 (1.2%) | |
| **Stress** | | | | | | <0.001 |
| Never | 1,235 (4.9%) | 107 (6.0%) | 81 (6.0%) | 639 (4.5%) | 408 (5.0%) | |
| Once | 8,962 (35%) | 582 (33%) | 480 (36%) | 5,044 (36%) | 2,856 (35%) | |
| Once the past five years | 9,881 (39%) | 656 (37%) | 520 (39%) | 5,486 (39%) | 3,219 (40%) | |
| Constant the past year | 2,631 (10%) | 207 (12%) | 113 (8.4%) | 1,435 (10%) | 876 (11%) | |
| Constant the past five years | 2,672 (11%) | 228 (13%) | 150 (11%) | 1,520 (11%) | 774 (9.5%) | |
| **Physical activity** | | | | | | <0.001 |
| Sedentary | 2,902 (11%) | 269 (15%) | 237 (18%) | 1,472 (10%) | 924 (11%) | |
| Moderate exercise | 11,811 (47%) | 811 (46%) | 628 (47%) | 6,725 (48%) | 3,647 (45%) | |
| Moderate but regular exercise | 7,628 (30%) | 491 (28%) | 348 (26%) | 4,337 (31%) | 2,452 (30%) | |
| Regular exercise and training | 3,040 (12%) | 209 (12%) | 131 (9.7%) | 1,590 (11%) | 1,110 (14%) | |
| **Sleep medication** | 357 (1.4%) | 31 (1.7%) | 26 (1.9%) | 212 (1.5%) | 88 (1.1%) | 0.011 |

1 Mean (SD); n (%)

2 Kruskal-Wallis rank sum test; Pearson's Chi-squared test

(Kruskal-Wallis rank sum test or chi2 test p < 0.005) (Table 1). In line with current studies [17], we found an observable discrepancy in coffee consumption between the two sexes (Fig 2A), with men showing a higher average coffee consumption compared to women. In comparison, age displayed a smaller influence on coffee consumption, although with statistically significant differences. In accordance with previous studies [18], there were significant differences in coffee consumption based on BMI, where individuals classified as underweight tend to drink less coffee compared to overweight individuals or those classified as obese (Fig 2A). Unlike coffee, tea shows a lower frequency of consumption among the participants, where about 35% report drinking tea 1 times a day or more. As previously reported [19,20], smoking status seems to influence coffee consumption, where individuals who smoke tend to drink on average more coffee than ex-smokers, and the difference is even more pronounced between current smokers and never-smokers.

## Habitual coffee consumption is associated with known genetic marks

To assess whether the questionnaire responses provided constitute reliable descriptors of coffee consumption, we investigated the presence of known genetic markers previously associated with coffee consumption, by performing a

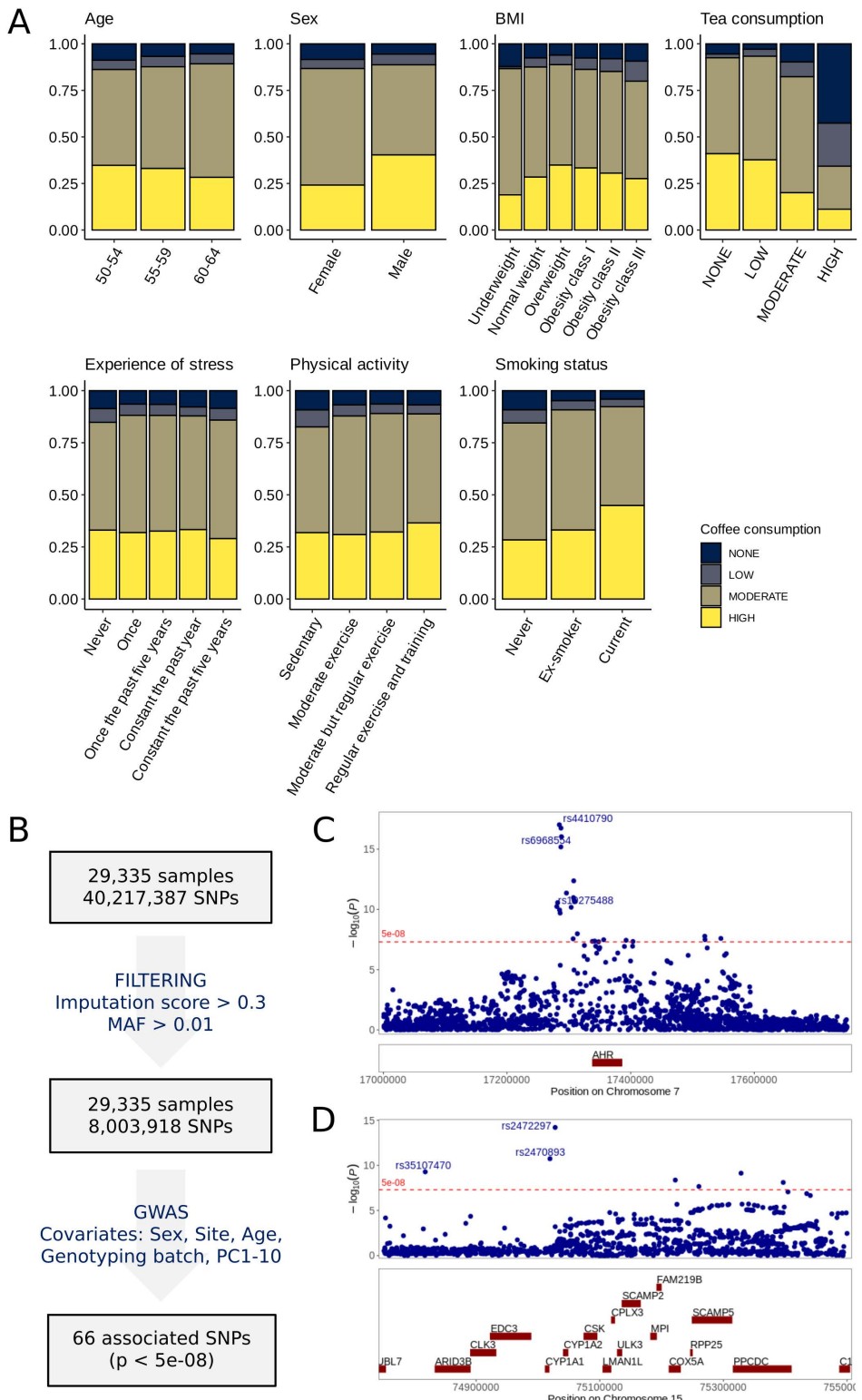

**Fig 2. Coffee consumption in the studied population and association with known genetic markers. (A)** Stacked bar plots representing the distribution of coffee consumption (y-axis) for each level (x-axis) of age, sex, BMI, tea consumption, stress, physical activity, and smoking status, respectively.

BMI has been divided into categories according to guidelines from the World Health Organization (WHO). **(B)** Simplified schematic description of key steps used in the genome-wide association studies (GWAS) analysis. **(C-D)** Region plots showing zoomed-in view at chromosome 7 **(C)**, and chromosome 15 **(D)**. Blue points represent individual SNPs. A p-value threshold of 5e-08 was used for determining significance (red-colored horizontal dashed line). Top significantly associated SNPs have been annotated in the plots.

genome-wide association study (GWAS) (Fig 2B). We identified 66 SNPs significantly associated with coffee consumption (p < 5e-8) (S2A Fig, S2 Table). 26 SNPs were located on chromosome 7p21 near the Aryl Hydrocarbon Receptor gene (*AHR*) with the top associated SNP rs4410790-T (Fig 2C), and 7 SNPs were located on chromosome 15q24 near the Cytochrome P450 Family 1 Subfamily A Member 1 and 2 genes (*CYP1A1*/*CYP1A2*) with the top SNP rs2472297-T (Fig 2D). An additional 33 SNPs were located on chromosome 22q11.23 near the Calcineurin Binding Protein 1 gene (*CABIN1*) and the Sushi Domain Containing 2 gene (*SUSD2*) with the top SNP rs6004089-C (S2B Fig). The SNPs on chromosomes 7 and 22 presented negative β-coefficient (also known as effect size estimate) indicating that the SNPs considered were negatively correlated with higher value of coffee consumption; the SNPs on chromosome 15, instead, had positive β-coefficient (S2 Fig C). The fact that these GWAS resulted in known genetic variants previously found to be associated with coffee consumption constitutes a powerful orthogonal confirmation of the robust reliability of our questionnaire-based data.

## Coffee intake shows low association with subjective estimates of sleep or daytime sleepiness

We examined the association between the distribution of coffee consumption described and validated above with i) sleep pattern/quality and ii) subjective estimate of daytime sleepiness. To measure sleep pattern/quality we evaluated subjective factors, reported by our cohort of participants, that are known objective indicators of sleep quality. These are: hours of sleep, quality of sleep, difficulty of falling asleep, waking up at night, waking up too early, reflux after going to bed, and loud snoring (S3 Table). In addition to considering each of these sleep factors separately, we devised a sleep score that combines them (see method section). The overall distribution of sleep scores was skewed towards lower values (mean = 8.6, SD = 4.47), indicating that most participants had a good sleep quality in general (Fig 3A, left panel). Likewise, and supporting the use of this metric, the distribution of the commonly used Epworth Sleepiness Scale (ESS) scores was also skewed towards lower values (mean = 6.3, SD = 4.14), indicating that most participants in the cohort experienced low overall daytime sleepiness (Fig 3A, right panel). Only 16% (4070/25381) of participants had an ESS score above 10 (the typical threshold for excessive sleepiness).

We found a quantitatively small, although statistically significant due to the large sample size, correlation between coffee consumption and sleep score (Kendall's tau = −0.045, p < 0.001) or ESS score (Kendall's tau = 0.011, p = 0.016). To assess whether this low correlation is due to an underlying relationship rather than bias in the data, we performed regression analysis (see details in method section), using each of the sleep factors, sleep score and ESS score as outcomes, respectively, and coffee consumption as main predictor, with adjustment for confounders identified by the DAG analysis. Statistical tests indicated possible violation of the proportional odds assumption, but this only translated to small differences in predicted category probabilities (min: 0.0002, 1st quantile: 0.0086, median: 0.0187, mean: 0.0233, 3rd quantile: 0.0359, max: 0.0766) (S3 Fig). While sensitivity analyses did imply some loss of information due to our aggregation of coffee consumption groups (S4 Table), visual inspection of predicted probabilities suggests non-linear patterns to be mild and still capturable by the current coffee consumption grouping (S4 Fig).

Counterintuitively, all sleep factors but sleep time and frequency of snoring indicated better sleep with higher coffee consumption (Fig 3B). Specifically, low coffee consumption was associated with worse sleep quality (Odds Ratio = 1.16, p = 0.023), higher frequency of difficulty falling asleep (Odds Ratio = 1.17, p = 0.018) and higher frequency of waking up during the night (Odds Ratio = 1.17, p = 0.014), compared to non-drinkers. This relationship was reversed with increased coffee consumption, where high consumption was associated with better sleep quality (Odds Ratio = 0.83, p < 0.001) and

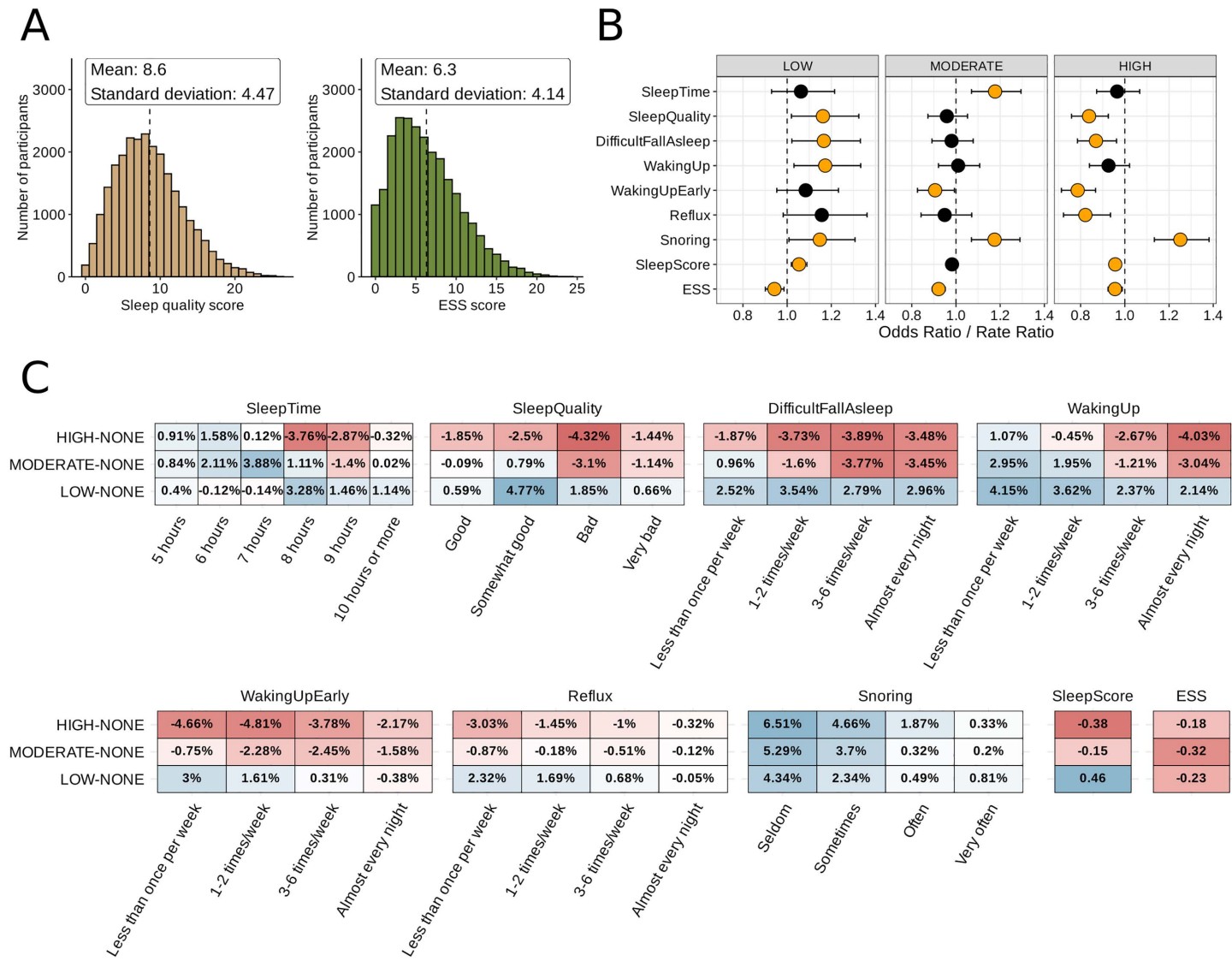

**Fig 3. Influence of habitual coffee consumption on sleep and day-time sleepiness. (A)** Histogram showing the distribution of calculated sleep scores (left panel) and ESS scores (right panel) across participants. Vertical dashed line represents the mean value. **(B)** Association of habitual coffee intake assessed by regression analysis. For each of the seven sleep factors, an ordinal logistic regression model was fitted, while for each of the sleep score and ESS score, a Quasi-Poisson regression model was fitted. All models were adjusted for basic demographics (age and sex), BMI, and lifestyle factors (habitual tea intake, stress, physical activity, and smoking), as identified by the DAG analysis. Dots show cumulative odds ratio (ordinal logistic regression) or rate ratio (Quasi-Poisson regression). Black lines on each side of the dots represent 95% confidence interval. Vertical dashed line shows Odds Ratio = 1. Significant results ($p < 0.05$) are highlighted as yellow dots. **(C)** Heatmaps showing differences in predicted outcomes. For categorical sleep factors (sleep duration, sleep quality, difficulty falling asleep, frequency waking up, frequency waking up too early, reflux, and snoring) partial pro-portional models were used to calculate predicted category probabilities. For sleep score and ESS score, quasi-Poisson models were used to calculate predicted counts. Results are presented as the difference in predicted probabilities/counts comparing coffee consumption level (high vs none, moderate vs none, and low vs none, respectively).

less difficulty falling asleep (Odds Ratio = 0.86, $p = 0.005$), and slightly (non-significant) lower frequency of waking up during the night (Odds Ratio = 0.92, $p = 0.12$). Likewise, high coffee consumption was associated with lower frequency of waking up to early (Odds Ratio = 0.78, $p < 0.001$) and lower frequency of reflux after going to bed (Odds Ratio = 0.82, $p = 0.003$). All

levels of coffee consumption were associated with higher frequency of loud snoring (Odds Ratios: 1.15–1.25, p: 0.037 –<0.001), with a slight positive trend with increased coffee consumption. Lastly, there was a slight association between higher coffee consumption and lower overall sleep score, while all levels of coffee consumption were associated with slightly lower ESS score with no apparent trend across the consumption levels.

However, while the association between coffee consumption and sleep factors were statistically significant, it translated to very small changes in actual probability distribution. Across the seven sleep variables we studied, absolute differences in predicted category probabilities between coffee consumption groups were small overall (median 1.9%, IQR 0.9–3.6%, maximum 7.7%). For Sleep score and ESS the differences in predicted counts between coffee consumption groups were also very small (minimum −0.84, median −0.2, IQR −0.33-(−0.05), maximum 0.46). (Fig 3C, S5 Table). This limited impact of coffee consumption on sleep factors is further highlighted when examining the importance of each included variable in the regression model (S5 Fig), where factors such as BMI and stress exhibit stronger explanatory power. Lastly, interaction analysis indicated improved model fit for interaction between coffee consumption and age in relation to ESS score (p = 0.033), interaction between coffee consumption and sex in relation to frequency of waking up (p = 0.045), and interaction between coffee consumption and BMI in relation to difficulty falling asleep (p = 0.002) (S6A Fig). However, and in line with the variable importance results, only BMI exhibited clear changes in predicted probability, where low coffee consumption was associated with increased probability of difficulty falling asleep with increased BMI (S6B-S6D Fig).

## Discussion

### Reported coffee consumption and genetic validation

Based on the collected data, a majority of study subject (87.7%) frequently drink coffee (>1 times/day), consistent with the overall high prevalence of coffee consumption in Sweden [15]. While these are self-reported data, which is commonly influenced by bias and uncertainties [21], GWAS supports the reliability of our data, as evident by the identified known variants at the *AHR* and *CYP1A1/2* genes.

In addition to the well-established coffee-associated genetic markers at the genes *AHR* and *CYP1A1/CYP1A2*, significant association was also observed for 33 SNPs at chromosome 22, of which several were located within the genes *CABIN1* and *SUSD2*. *CABIN1* encodes a nucleus-located protein that binds specifically to activated calcineurin and inhibits calcineurin-mediated signal transduction [22]. While there is no previously known connection between genetic variants in the *CABIN1* gene and coffee consumption, studies have shown that calcineurin and its regulation is required for normal sleep in Drosophila [23,24], and for sleep regulation in mammals [25]. Although this is an interesting finding, we choose not to elaborate on it here but propose further studies to determine whether genetic variation at *CABIN1/SUSD1* has any relevant role in susceptibility to the effects of caffeine.

Another locus of importance is the Adenosine A2a receptor gene (*ADORA2A*), in which genetic variation has been associated with individual sensitivity to caffeine effects on sleep [26]. However, while a collection of SNPs exhibited a higher association around the *ADORA2A* gene in our dataset, none met the threshold for significance (p < 5e-8). This lack of statistically significant association may be explained by the relatively high age of the study population. Given the previously demonstrated adaptive capacity of the adenosine system at prolonged caffeine exposure, it is possible that coffee consumers within the studied group are less susceptible to the effect of caffeine, thus masking the potential importance of genetic variation at the *ADORA2A* locus.

### What is the effect of coffee consumption on sleep/sleepiness?

Contrary to the general perception of the wakefulness-inducing effects of caffeinated beverages, such as coffee, our results suggest that the correlation between habitual coffee consumption level and perceived sleep or daytime sleepiness is negligible. While our regression analyses indicate some statistically significant relationships between coffee

consumption and sleep/sleepiness, these correspond to very low predicted effect sizes, with likely limited physiological impact. Our analysis did, however, reveal BMI as a potential contributing factor on the effect of coffee consumption on reported frequency of difficulty falling asleep at night. This is supported by a recent study in adult women which reports prolonged and more adverse caffeine effects in obese subjects compared to non-obese [27]. Our results thus support the need to consider individual body composition in caffeine dose recommendations.

The unexpected lack of strong overall relationship between coffee consumption and perceived sleep quality or daytime sleepiness could be due to a recently noted adaptive feature of the adenosine system [4]. While acute caffeine administration, especially in the evening close to bedtime, has shown effects on sleep efficiency and latency, study evidence suggests that these effects normalize during prolonged daily caffeine intake [28,29]. This has also been observed on subjective daytime sleepiness, as self-reported by patients. Furthermore, in the mouse, results have shown that repeated caffeine exposure leads to an adaptive response in the brain characterized by increased levels of adenosine [30] and an increase in the number of adenosine receptors [31–34]. Thus, as our data set constitutes information on individuals within the age range of 50–64, this may explain the absence of a stronger effect of coffee consumption on perceived sleep. Those who drink coffee have probably done so for many years, as we assume it is unlikely that a stable habit of coffee consumption is initiated later in life. The adenosine system of these individuals could therefore already have experienced an adaptation to chronic caffeine exposure. Of note, recent studies have demonstrated reduced levels of adenosine receptor $A_1$ and overexpression of $A_2$ in the brain of aging rats [35] and aging humans [36], raising the question of the role of age in the response to caffeine.

### Study limitations

While the participants estimated the frequency of coffee consumption according to pre-defined categories, no data was collected on the number of cups, the type of coffee beans, caffeine concentration, or whether the participants consume decaffeinated coffee. It is also possible that participants may consume caffeine from sources other than coffee or tea, such as energy drinks or certain soft drinks. Data on the consumption of such drinks was not included in this study. However, although we cannot rule out some effects of these neglected caffeine sources, it is likely that coffee and tea are the main sources, as their consumption tends to have a more consistent and habitual pattern. In addition, the timing of coffee consumption has not been considered in the analysis. While caffeine has been shown to reach near-complete absorption within 45 minutes and with a mean elimination half-life of about 5 hours [37,38], we cannot rule out the possibility that participants consume coffee close to bedtime, within the average half-life of caffeine where an effect on sleep could be plausible. Medical comorbidities, yet another potential confounder that could influence the associations, have not been accounted for as this information has not been readily available for analysis. Furthermore, as this study was conducted at northern latitudes, it is possible that seasonal differences could affect estimated coffee consumption or subjective estimates of sleep or sleepiness. In summary, while possible confounders have been accounted for in our analysis, we cannot exclude that there may be additional confounders. Sleep and sleepiness are complex behavioral traits that are influenced by a large variety of factors [39], and it is therefore likely that coffee consumption and sleep co-vary as a function of several confounding variables.

Lastly, it is important to note that the cross-sectional design of this study, while providing descriptive associations, does not fully permit causal inference. Thus, further studies are needed to gain a definitive picture of the long-term effects of caffeine intake on sleep and sleepiness.

### Conclusions

The present study indicates that habitual coffee consumption, in a middle-aged Swedish cohort, show no, or at least only a weak, association with estimates of sleep quality and perceived daytime sleepiness. This lack of a clear association could be due to the older age of the study participants, in whom prolonged exposure to caffeine probably has led to a normalization

of the adenosine system. A possible explanation could lie in the adaptability of the adenosine system. However, we believe that further studies, with larger populations or comparative analyses between older and younger cohorts, are needed to fully ascertain the long-term effects of coffee on sleep and sleepiness, and the role of adenosine adaptation and/or ageing in the response to caffeine exposure.

## Methods

The following is a summary of the main methods used in the study. Additional methods details can be found in supplementary materials.

### Ethics approval and consent to participate

Ethical approval was obtained from the Swedish Ethical Review Board (reference number 2023-01559-01). The SCAPIS study was conducted in accordance with the Declaration of Helsinki, and the study has been approved as a multicenter trial by the Ethics committee at Umeå University (Dnr. 2010–228-31M). All participants provided written informed consent.

### Data selection

From the main SCAPIS dataset (accession gained at 16/08/2023), samples which had genotyping results were selected, corresponding to a total of 29,335 individuals (14,997 females and 14,338 males). This subset was used for genetic analyses (see further details below and in supplementary materials).

For information on coffee consumption, participants answered a food frequency questionnaire [40]. Possible answers were "Not applicable" (which corresponds to no coffee consumption), "1-3 times/month", "1-2 times/week", "3-4 times/ week", "5-6 times/week", "1 time/day", "2 times/day", "3 times/day", "4 times/day", and "5 times/day or more". Based on these answers, the participants were further divided into four groups: None, Low (1time/month – 6time/week), Moderate (1–3 times/day), and High (4–5 times/day or more) coffee consumption. This grouping was chosen to ensure sufficient observations per category and stable model estimation.

Information on different aspects of sleep has been reported from a self-administered questionnaire on sleep habits, adapted from the Basic Nordic Sleep Questionnaire [41]. A total of seven sleep-related factors were selected for analysis, including hours of sleep, sleep quality, difficulty falling asleep, frequency of waking up several times, frequency of waking up too early, reflux after going to bed, and frequency of loud snoring. The participants were rating these, except for hours of sleep, on a 5-point scale (0–4) from good/low frequency to bad/high frequency.

Since each of the sleep factors constitutes different aspects of sleep, we also considered them together, by calculating an overall sleep quality score, defined as the sum of the individual sleep factors. Since hours of sleep were reported on a different scale, we recoded it as follows: 7 or more hours of sleep was given a score of 0, 6 hours of sleep was given a score of 1, 5 hours of sleep was given a score of 2, and 4 hours or less was given a score of 3. The resulting sleep score ranges from 0 to 27, with higher values corresponding to wors overall sleep. A similar sleep health score has been used in a previous study, although in the context of cardio-vascular disease [42,43].

In addition to nighttime sleep, participants were also asked to rate, on a 4-point scale (0–3), how likely they are to doze off or fall asleep during the day while engaged in 8 different activities. This method of assessing daytime sleepiness (DS) is known as the Epworth Sleepiness Scale (ESS), developed by Dr. Johns in 1990 [44]. A total ESS score is obtained from the sum of rated scores from the 8 activities, ranging from 0 to 24. General interpretation of the ESS scores has been suggested by the inventor of the method, in which a score of 0–5 indicates lower normal DS, 6–10 indicates higher normal DS, 11–12 indicates mild excessive DS, 13–15 indicates moderate excessive DS, and 16–24 indicates severe excessive DS.

 

## Missing data and imputation

Patterns in missing data were assessed by Little's MCAR test using the R package naniar (version 1.1.0) [45]. Missing-ness was also further investigated by analyzing how missing data correlated between variables (Pearson correlation). To investigate the reliability of missing data imputation, multiple imputation by multiple correspondence analysis (MCA) were performed using the R package missMDA (version 1.19) [46]. The robustness of the imputation was assessed through an overimputation procedure, in which observed data points were randomly masked (corresponding to 10% of data) and imputed (100 iterations). The overimputed data were then compared to the true observed values to evaluate the accuracy of imputation. To contextualize this performance, we also compared it to the expected accuracy under random guessing based on marginal category frequencies ("naive baseline") and expressed this as a ratio.

## Directed acyclic graph (DAG) analysis

Both coffee consumption and sleep/sleepiness are complex human behavioral traits and may be influenced by several other confounding factors. Therefore, a directed acyclic graph (DAG) analysis was conducted to identify such relevant confounding factors. Nighttime sleep and daytime sleepiness, while distinct aspects of a person's life, are heavily interconnected. Therefore, we defined a single DAG for the relationship between coffee consumption and sleep/ESS.

## Data filtering

Participants with missing data in at least one of the sleep/ESS variables were removed, as well as those with missing data on coffee consumption or any of the confounders identified by the DAG analysis, leaving a total of 25,381 participants that were included in subsequent analyses. Imputation of missing data was not considered suitable in this case.

## Main statistical analysis

Statistical analyses were conducted using R (version 4.4.2). Comparison of demographic and lifestyle characteristics between coffee consumption groups was performed with Kruskal-Wallis rank sum test for continuous variables, and Pearson's Chi-square test for categorical variables. Correlation analyses were assessed by calculating the Kendall rank correlation. A p-value $< 0.05$ was used as threshold for significance.

The relationship between coffee consumption and sleep/ESS was assessed by regression modelling. For each of the sleep variables, treated as ordinal categorical outcome variables, a cumulative link model was fitted using the R package ordinal [47] (version 2023.12.4.1), with coffee consumption as main predictor, and with adjustment for confounding factors as identified by the DAG analysis (additional details in supplementary materials). Results were presented as odds ratios and corresponding p-values. Sleep score and ESS score were modeled using quasi-Poisson generalized linear models, with coffee consumption as main predictor and with adjustment for confounding factors as identified by the DAG analysis. Results were presented as rate ratios and corresponding p-values. When modeling ESS score, sleep time was added as an additional confounder in the model. Variable importance for each model was summarized using analysis of variance (ANOVA) and Type II Wald Chi-square statistics. Predicted individual outcome probabilities were calculated and visualized using the R package ggeffects [48] (version 2.2.1), using partial proportional odds models, and presented as signed absolute risk differences. Potential effect modifications were investigated for by iteratively introducing an interaction term between coffee consumption and each of sex, age and BMI. We then assessed whether the inclusion of these interaction terms improved model fit compared to the original models, by using likelihood ratio tests for ordinal models and F-tests for the quasi-Poisson models. For any significant improvement of model fit, model-based estimates were calculated (adjusted predicted probabilities for ordinal models and adjusted predicted counts for the quasi-Poisson models).

## Diagnostics and sensitivity analyses

To ensure that the aggregation of coffee consumption groups do not obscure non-linear dose-dependent patterns in data, a sensitivity analysis was conducted in which, for each outcome, four different models were fitted: Model 1 (M1) with coffee consumption as a four-category variable, Model 2 (M2) with coffee consumption as the original questionnaire categories, Model 3 (M3) with coffee consumption modeled as a continuous variable using numeric approximate midpoints for each outcome category under the assumption of a linear association, and Model 4 (M4) with coffee consumption modeled continuously using restricted cubic splines to allow for non-linear associations. Numeric midpoints were assigned to the frequency categories to approximate average consumption levels (e.g., weekly and daily frequencies converted to times/day). This approach enabled formal testing of dose–response trends while accounting for unequal category spacing. Nested models were compared using likelihood ratio tests for ordinal outcomes and F-tests for count outcomes. Specifically, M1 was compared with M2 to evaluate potential information loss due to category aggregation, and M3 was compared with M4 to assess evidence of non-linearity. Model fit was further evaluated using Akaike Information Criterion (AIC) and Bayesian Information Criterion (BIC) for likelihood-based models. For quasi-Poisson models, a quasi-AIC (QAIC) was calculated. Pseudo $R^2$ measures and model deviance were also extracted to facilitate comparison across models. Predicted outcome values across the range of coffee consumption were visualized using marginal effect plots derived from the fitted models. These plots were used to support interpretation of linear versus non-linear associations.

For cumulative link models fitted for ordinal categorical variables, the proportional odds assumption (PO) was assessed by nominal test and scale test. To quantify any possible PO violation, we fitted partial proportional odds models, using the same formulas and covariates as the main analysis, and computed threshold-specific odds ratios and cumulative logit, as well as predicted category probabilities.

## Genome-wide association studies (GWAS)

GWAS was performed on filtered genotype data (info score > 0.3, mean allele frequency (MAF) > 1%) to identify SNPs associated with coffee consumption. A p-value < 5e-08 was used as threshold for significance. Sex, age, genotyping batch and the first ten principal components were included as covariates. Analysis was performed using PLINK 2.0 [49] and visualization of GWAS results were made using the R package topr 2.0.0 [50].

## Supporting information

**S1 Fig. Data missingness analysis.** (A) Heatmap showing missingness patterns for each variable (x-axis) across coffee consumption levels (y-axis) before aggregation of coffee consumption into fewer groups. Values in each cell corresponds to percentage of missing values. (B) Heatmap showing Pearson correlation of missing values between all variables. The color map corresponds to the Pearson correlation coefficient. (C) Schematic overview of the procedure to assess the efficiency and accuracy of the data imputation, using an overimputation procedure. (D) Horizontal bar plot showing the mean accuracy of overimputed values. A value of 1 corresponds to perfect accuracy. (E) Horizontal bar plot showing the ratio between overimputation accuracy and random guessing. Values close to 1 indicate that that the imputation is not better than random guessing.
(TIF)

**S2 Fig. Additional GWAS results overview and associations at chromosome 22.** (A) Manhattan plot showing the p-values of the entire GWAS on a genomic scale, for coffee consumption. Blue points represent individual SNPs tested in the analysis. A p-value threshold of 5e-08 was used for determining significance (red-colored horizontal dashed line). (B) Region plot showing zoomed-in view at chromosome 22. Blue points represent individual SNPs. A p-value threshold of 5e-08 was used for determining significance (red-colored horizontal dashed line). Top significantly associated SNPs at

each of the identified loci have been annotated in the plots. (C) Horizontal lollipop plot showing effect sizes (beta value) for each significant SNP.
(TIF)

**S3 Fig. Proportional odds assumption diagnostics checks.** (A) Nominal test and scale test assessing the proportional odds assumption (PO). The heatmap shows the statistical significance obtained by these two tests for each of the modeled sleep traits (y-axis) and each predictor/covariate included in the models (x-axis). Color map corresponds to p-value. *=p<0.05, **=p<0.001, ***=p<0.0001. (B) Predicted cumulative logits for each coffee group plotted across the thresholds of each sleep variable. A roughly constant vertical separation between coffee groups across thresholds indicates that the PO holds in logit-space. In the present analysis, some changes in separation are observed, pointing to a possible violation of PO. (C) Threshold-specific odds ratios (ORs) obtained by fitting a partial proportional odds model. A large spread in ORs would indicate statistical non-proportionality. I the present analysis, ORs follow similar patterns. (D) Predicted category probabilities (y-axis) across coffee consumption groups (x-axis). Categories of each sleep variable are color-coded (see figure legend). Predicted-probability differences are small across categories, indicating that the PO violation is not practically important.
(TIF)

**S4 Fig. Sensitivity analysis investigating non-linear patterns in data.** Plots showing predicted category probabilities or predicted counts. Four different models designs were tested: with coffee consumption divided by the original 10 questionnaire categories (leftmost column), with coffee consumption divided by four aggregated categories (second column from the left), with coffee consumption as a continuous variable based on numeric approximate midpoints for each outcome category (second column from the right), and with coffee consumption modeled continuously using restricted cubic splines (rightmost column). For categorical sleep outcomes, cumulative link models were fitted (leftmost and second to left columns, first seven rows). For count-like sleep score and ESS, quasi-Poisson generalized linear models were fitted (rightmost and second to right columns, last two rows).
(TIF)

**S5 Fig. Variable importance in regression models.** Variable importance of each factor (y-axis) for each fitted regression model (x-axis), assessed by analysis of variance (ANOVA). *=p<0.05, **=p<0.001, ***=p<0.0001. Color code represents the Type II Wald Chi-square statistics, where higher value (darker color) corresponds to a stronger contribution of the variable to the model.
(TIF)

**S6 Fig. Interactions analyses.** Investigation of potential effect modification by sex, age and BMI. (A) Assessment of improved model fit when introducing an interaction term between coffee consumption and age, sex and BMI, respectively (x-axis). This was performed for each of the outcomes studied (y-axis). For ordinal models, a likelihood ratio test was used. For quasi-Poisson models, an F-test was used. Significant results (p<0.05) are highlighted in blue. (B) Adjusted predicted category probabilities (y-axis) showing the effect of coffee consumption level on difficulty to fall asleep, stratified by BMI (x-axis). (C) Adjusted predicted counts (y-axis) showing the effect of coffee consumption level on ESS score, stratified by age (x-axis). (D) Adjusted predicted category probabilities (y-axis) showing the effect of coffee consumption level (x-axis) on frequency waking up during night, stratified by sex.
(TIF)

**S1 Table. Details on each SCAPIS variable selected for the present study.** Details on each SCAPIS variable selected for the present study, including method of data collection, questionnaire question wording and possible answers, and how the variable was used in the present study.
(DOCX)

**S2 Table. Significant associated SNPs obtained by GWAS on coffee consumption.** Columns from left to right: name of the SNP, the chromosome on which the SNP is located, the single nucleotide position of the SNP, the allele that was tested in the analysis, the alternative allele, mean allele frequency of the tested allele, the effect size estimate (beta) of the SNP, the standard error (se) of the effect size estimate, and the statistical significance (p-value) for the association. (DOCX)

**S3 Table. Patterns of habitual coffee consumption by sleep factors and ESS.** (DOCX)

**S4 Table. Sensitivity analysis results for assessing non-linearity.** Four models were fitted: Model 1 (M1) with coffee consumption as a four-category variable, Model 2 (M2) with coffee consumption as the original questionnaire categories, Model 3 (M3) with coffee consumption modeled as a continuous variable using numeric approximate midpoints for each outcome category under the assumption of a linear association, and Model 4 (M4) with coffee consumption modeled continuously using restricted cubic splines to allow for non-linear associations. M1 and M2 were compared to test whether collapsing the original coffee consumption categories into 4 groups causes meaningful information loss. M3 and M4 were compared to assess whether a model that allows non-linearity fits better to the data. Additional model metrics were also calculated, including Akaike Information Criterion (AIC), Bayesian Information Criterion (BIC), and Nagelkerke pseudo R-squared (R2). For quasi-Poisson models, Quasi-Akaike Information Criterion (QAIC) and deviance are given. (DOCX)

**S5 Table. Adjusted predicted probabilities.** Adjusted predicted probabilities for sleep categories based on coffee consumption level and signed absolute differences between coffee consumption groups. For sleep score and ESS score, values are given as predicted counts. (DOCX)

**S1 File. Additional methods details.** (PDF)

## Author contributions

**Conceptualization:** Simon Söderholm, Martin Ulander, Claudio Cantù, Fredrik Iredahl.

**Data curation:** Simon Söderholm.

**Formal analysis:** Simon Söderholm.

**Funding acquisition:** Fredrik Iredahl.

**Investigation:** Simon Söderholm.

**Methodology:** Simon Söderholm.

**Project administration:** Simon Söderholm, Claudio Cantù, Fredrik Iredahl.

**Resources:** Daniel Berglind, Susanna Calling, Bledar Daka, Ludger Grote, Mats Martinell, Frida Bergman, Pontus Henriksson, Carl-Johan Östgren, Claudio Cantù, Fredrik Iredahl.

**Software:** Simon Söderholm.

**Supervision:** Claudio Cantù, Fredrik Iredahl.

**Validation:** Simon Söderholm, Claudio Cantù, Fredrik Iredahl.

**Visualization:** Simon Söderholm.

**Writing – original draft:** Simon Söderholm, Claudio Cantù, Fredrik Iredahl.

**Writing – review & editing:** Simon Söderholm, Martin Ulander, Vanessa William Toma, Sara Kaufmann, Xiangyu Qiao, Daniel Berglind, Susanna Calling, Bledar Daka, Ludger Grote, Mats Martinell, Frida Bergman, Pontus Henriksson, Carl-Johan Östgren, Wen Zhong, Claudio Cantù, Fredrik Iredahl.

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
