## [Decision Letter · Decision Letter 0]

3 Oct 2025

Dear Dr. Söderholm,

Thank you for submitting your manuscript to PLOS ONE. After careful consideration, we feel that it has merit but does not fully meet PLOS ONE’s publication criteria as it currently stands. Therefore, we invite you to submit a revised version of the manuscript that addresses the points raised during the review process.

We look forward to receiving your revised manuscript.

Kind regards,

Julio Alejandro Henriques Castro da Costa

Academic Editor

PLOS ONE

[C.C is a fellow and F.I is an associated clinical fellow of Wallenberg Center for Molecular Medicine (WCMM) and receives financial support from the Knut and Alice Wallenberg Foundation. C.C is also supported by funding from the Swedish Research Council, Vetenskapsrådet (2021-03075 and 2023-01898) and Cancerfonden (CAN 2018/542 and 21 1572 Pj). This work was also financed by Dr P Håkanssons Foundation, and the SciLifeLab & Wallenberg Data Driven Life Science Program (KAW 2020.0239).].

Additional Editor Comments (if provided):

Reviewers' comments:

Reviewer's Responses to Questions

**Comments to the Author**

1. Is the manuscript technically sound, and do the data support the conclusions?

Reviewer #1: Partly

Reviewer #2: Yes

2. Has the statistical analysis been performed appropriately and rigorously?

Reviewer #1: No

Reviewer #2: I Don't Know

3. Have the authors made all data underlying the findings in their manuscript fully available?

Reviewer #1: No

Reviewer #2: No

4. Is the manuscript presented in an intelligible fashion and written in standard English?

Reviewer #1: Yes

Reviewer #2: Yes

Reviewer #1: Dear Editor and authors,

Thank you for the opportunity to review this manuscript. The authors investigate the association between habitual coffee consumption and sleep quality/daytime sleepiness in a large Swedish cohort (SCAPIS, n≈25,000), complemented by genetic validation of self-reported coffee intake. The topic is timely and relevant, given the widespread consumption of caffeine and its disputed long-term effects on sleep. The sample size and the genetic analysis are strengths. However, several methodological and reporting issues limit the interpretability and impact of the study. Below I provide my detailed comments.

Mayor concerns:

1. Study design and causal inference

The cross-sectional design is suitable for descriptive associations but inherently limited for causal inference. The discussion should more explicitly acknowledge that the analyses cannot establish long-term effects of coffee on sleep or sleepiness.

2. Handling of missing data

The authors excluded all participants with any missing data (complete-case analysis). This approach may introduce selection bias if data are not missing completely at random. A description of the missingness patterns and, ideally, multiple imputation or sensitivity analyses are required.

3. Exposure measurement (coffee consumption)

Coffee intake was captured through a food frequency questionnaire and grouped into four categories. This grouping collapses possible non-linear associations (e.g., U-shaped curves) and may obscure dose–response effects.

The “high” group is labeled as 4–5 cups/day, but it seems to include “≥5/day” as well; please clarify.

No information is available on decaffeinated coffee, timing of consumption (morning vs. evening), or other caffeine sources (energy drinks, sodas). These are important determinants of caffeine’s effect on sleep and should at least be discussed.

4. Sleep variables: description and use

Seven items were collected (hours of sleep, sleep quality, sleep latency, nocturnal awakenings, early awakening, reflux, loud snoring). Most were rated on 0–4 Likert scales, but “hours of sleep” was recorded on a different scale.

The authors constructed a composite “sleep score” summing six items and *excluded hours of sleep*. They justify this on grounds of different scaling and considering duration as preference rather than quality. While defensible, sleep duration is a central dimension of sleep health and strongly linked to sleepiness. Excluding it weakens the construct validity of the score.

For robustness, I strongly recommend sensitivity analyses that incorporate hours of sleep (after standardization), or alternatively provide two scores: “sleep quality” (without duration) and “sleep health” (with duration).

Furthermore, the methods should describe how hours of sleep were categorized or treated in models. This is insufficiently specified.

5. Statistical modeling

Use of cumulative link models for ordinal sleep variables and quasi-Poisson regression for ESS and the composite score is appropriate. However, the manuscript does not report diagnostic checks, particularly the proportional odds assumption in ordinal models. This is a major omission.

ESS is bounded (0–24) and often skewed. Modeling it as quasi-Poisson is unusual; linear robust regression or negative binomial could be tested in sensitivity analyses. Categorizing ESS (≤10, 11–12, ≥13) might also aid interpretability.

Multiple comparisons: with several sleep factors and models tested, there is a risk of type I error inflation. The authors should clarify whether they applied any correction or provide justification for not doing so.

6. Clinical relevance versus statistical significance

With n>25,000, even trivial associations reached p<0.05 (e.g., Kendall’s tau = –0.045). The manuscript must emphasize that effect sizes are extremely small and not clinically meaningful. Reporting standardized effect sizes and absolute risk differences would improve clarity.

7. Confounding and covariates

The DAG approach to select confounders is commendable. However, important potential confounders were omitted, notably medical comorbidities and use of sleep medications. These could substantially bias associations and should be at least acknowledged more fully.

Interaction analyses (e.g., by sex, age, BMI) would be informative.

8. Genetic analysis

The GWAS replicated known loci (AHR, CYP1A1/2), which validates self-reported coffee intake. The stratification by rs762551 (fast vs. slow metabolizers) is interesting but effect sizes were marginal.

The earlier Mendelian randomization approach has been removed, which is reasonable given prior concerns, but this reduces causal inference. The discussion should acknowledge this explicitly.

9. Data transparency and reproducibility

Data access requires approvals from SCAPIS, which is understandable, but limits reproducibility. The authors should at least ensure that full analysis code and derived data tables are made openly available.

minor concerns:

1. Tables are dense and difficult to interpret; simplifying presentation or providing supplementary stratified tables would help.

2. The discussion repeats background literature without sufficient integration into the present findings. More emphasis on implications for clinical practice and public health is encouraged.

3. Figures could be streamlined; currently some are overly complex for the message conveyed.

4. The manuscript would benefit from clearer differentiation between “sleep quality” and “sleep health,” to avoid confusion for readers.

The manuscript addresses a relevant question with a strong dataset, but the methodology and interpretation require substantial improvement before being suitable for publication in a high-impact journal.

Reviewer #2: Given the significant limitations in the study (and the limitations in interpretation of the title terms "sleep quality and daytime sleepiness"), the title might be amended to include novel findings in the caffeine-associated GWAS that might stimulate further research.

Major limitation to use of term "sleepiness" in this paper. Sleepiness is the tendency to fall asleep and is measured as mean sleep latency. The correlation between the Epworth Sleepiness Scale and mean sleep latency is relatively poor with r=0.37 (Chervin et al. 1997). Use of this scale as a quantitative measure of sleepiness is problematic. Better to use the term, "subjective estimate of sleepiness."

The sleep quality score has not been validated against objective measurement of sleep quality, as implied. Reference 41, for validation of the "sleep quality score," derived from the Basic Nordic Sleep Questionnaire, simply refers to reference 40 for validation and can be omitted. It should be noted that in reference 40, the validation of the "sleep quality score" is for cardiovascular disease and it has not been validated with polysomnography as a measure of quality of sleep.

Accuracy of a subject's perceptions of sleep onset and duration can vary widely with actual measurement. As the authors note, caffeine has an effect on cognitive function and therefore, not surprisingly, caffeine consumption distorts one's perception of sleep and sleepiness (e.g., Schlichtiger et al., 2025). Effects of caffeine seem to further confound the already loose relationships among caffeine consumption, subjective estimates of sleepiness and sleep quality (questionnaires), and objective measurements of sleepiness and sleep quality (polysomnography).

The manuscript should be revised to reflect these limitations. I.e., "Coffee intake shows low association with [subjective estimates] of sleep or daytime sleepiness"

Similarlly - Line 166 "Likewise and confirming the validity of this metric...," should be softened to "supporting the validity...." or "supporting the use...."

At this northern latitude, did time of year significantly affect estimated coffee consumption or subjective estimates of sleepiness or sleep quality?

Line 189 - Was BMI a better correlate for loud snoring than coffee consumption? Is seasonal sleep and sleepiness and caffeine consumption another uncontrolled variable in interpretation?

Terms need to be sufficiently defined to eliminate the apparent discrepancy in conclusion of line 183 and line 191. 183 - "This relationship was reversed with increased coffee consumption, where high consumption was associated with better sleep quality..." and line 191 - "there was a slight association between higher coffee consumption and lower overall sleep score." (What is the difference between the terms "sleep quality" and "sleep score." The methods only define a "sleep quality score.") As written, these lines seem to be contradictory.

Line 201 - "Stress" correlates well with subjective estimates of sleep quality and sleepiness; perhaps the methods section could include more discussion of how this variable was determined.

Prior studies suggest slow metabolizers suffer more hyperglycemia, hypertension and MI. Further, the "Sleep quality score" has correlated with cardiovascular events.... I do not appreciate correlation among slow and fast metabolizers and usual sleep complaints associated with cardiovascular risk.

OVERALL, recommend de-emphasize negative results (for sleep and sleepiness) and emphasize novel (GWAS) findings.

**Do you want your identity to be public for this peer review?** For information about this choice, including consent withdrawal, please see our Privacy Policy

Reviewer #1: **Yes:** Carlos Ernesto Bolanos Almeid

Reviewer #2: No

---

## [Author Response · Author response to Decision Letter 1]

30 Jan 2026

We thank the editor and reviewers for their careful assessment of our work and for the opportunity to revise and improve our manuscript. Please find attached to the re-submission a separate file with our detailed response to all the editor's and reviewer's comments.

---

## [Decision Letter · Decision Letter 1]

23 Feb 2026

Habitual coffee consumption poorly correlates with sleep quality and daytime sleepiness: a cross-sectional study

PONE-D-25-44514R1

Dear Dr. Söderholm,

We’re pleased to inform you that your manuscript has been judged scientifically suitable for publication and will be formally accepted for publication once it meets all outstanding technical requirements.

Kind regards,

Julio Alejandro Henriques Castro da Costa

Academic Editor

PLOS One

Additional Editor Comments (optional):

Reviewers' comments:

Reviewer's Responses to Questions

**Comments to the Author**

Reviewer #1: All comments have been addressed

2. Is the manuscript technically sound, and do the data support the conclusions?

Reviewer #1: Yes

3. Has the statistical analysis been performed appropriately and rigorously?

Reviewer #1: Yes

4. Have the authors made all data underlying the findings in their manuscript fully available?

Reviewer #1: Yes

5. Is the manuscript presented in an intelligible fashion and written in standard English?

Reviewer #1: Yes

Reviewer #1: Thank you very much for the opportunity to evaluate and review the changes and adjustments made to the article; they seem correct and complete to me.

**Do you want your identity to be public for this peer review?** For information about this choice, including consent withdrawal, please see our Privacy Policy

Reviewer #1: **Yes:** Carlos Ernesto Bolanos-Almeida MD MsC

---

## [Editor Report · Acceptance letter]

PONE-D-25-44514R1

PLOS One

Dear Dr. Söderholm,

I'm pleased to inform you that your manuscript has been deemed suitable for publication in PLOS One. Congratulations! Your manuscript is now being handed over to our production team.

Kind regards,

on behalf of

Dr. Julio Alejandro Henriques Castro da Costa

Academic Editor

PLOS One